# Why Can't I Dance in the Mall?
# Learning to Mitigate Scene Bias
# in Action Recognition

**Jinwoo Choi**
Virginia Tech
jinchoi@vt.edu

**Chen Gao**
Virginia Tech
chengao@vt.edu

**Joseph C. E. Messou**
Virginia Tech
mejc2014@vt.edu

**Jia-Bin Huang**
Virginia Tech
jbhuang@vt.edu

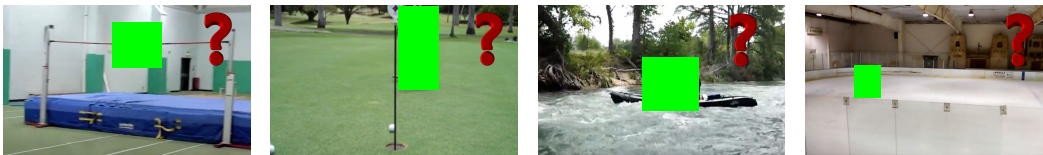

Figure 1: **Quiz time!** Can you guess what action the (blocked) person is doing in the four videos? Even though we cannot see a human actor, we can easily predict the action by considering where the scene is. Training a CNN model from these examples may lead to a strong bias toward recognizing the scene or the objects present in the video as opposed to paying attention to the actual action the person is doing. In this work, we show that learning video representation with debiasing leads to improved generalization to novel classes and tasks.

## Abstract

Human activities often occur in specific scene contexts, e.g., playing basketball on a basketball court. Training a model using existing video datasets thus inevitably captures and leverages such bias (instead of using the actual discriminative cues). The learned representation may not generalize well to new action classes or different tasks. In this paper, we propose to mitigate scene bias for video representation learning. Specifically, we augment the standard cross-entropy loss for action classification with 1) an adversarial loss for scene types and 2) a human mask confusion loss for videos where the human actors are masked out. These two losses encourage learning representations that are unable to predict the scene types and the correct actions when there is no evidence. We validate the effectiveness of our method by transferring our pre-trained model to three different tasks, including action classification, temporal localization, and spatio-temporal action detection. Our results show consistent improvement over the baseline model without debiasing.

## 1   Introduction

Convolutional neural networks (CNNs) [8, 64, 52, 60] have demonstrated impressive performance on action recognition datasets such as the Kinetics [31], UCF-101 [50], HMDB-51 [34], and others. These CNN models, however, may sometimes make correct predictions for wrong reasons, such as leveraging scene context or object information instead of focusing on actual human actions in videos. For example, CNN models may recognize a classroom or a whiteboard in an input video and predict the action in the video as *giving a lecture*, as opposed to paying attention to the actual activity in the scene, which could be, for example, *eating*, *jumping* or even *dancing*. Such biases are known as representation bias [38]. In this paper, we focus on mitigating the effect of *scene* representation

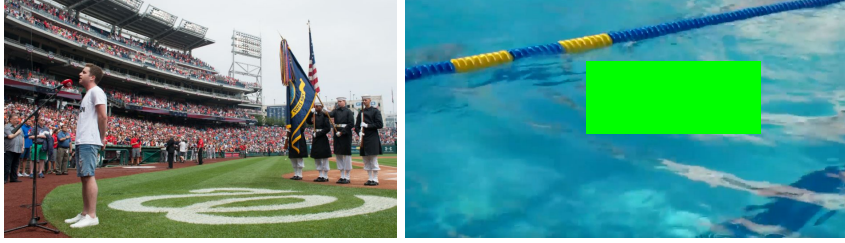

Figure 2: **Motivation of the proposed debiasing algorithm.** (*Left*): A man is ***singing*** in a baseball field. However, representations with certain bias toward the scene may predict the incorrect action e.g., , *playing baseball*. (*Right*): A person (masked-out) is ***swimming*** in a swimming pool. A model is capable of predicting the correct action i.e., *swimming* even without looking at the evidence. Video representations that make correct (or incorrect) predictions by leveraging the scene bias may not generalize well to unseen action classes and tasks.

bias [38]. Following Li et al. [38], we define the scene representation bias of a dataset $D$ as

$$B_{\text{scene}} = \log[M(D, \phi_{\text{scene}})/M_{\text{rand}}]. \tag{1}$$

Here, $\phi_{\text{scene}}$ is a scene representation; $M(D, \phi_{\text{scene}})$ is an action classification accuracy with a scene representation $\phi_{\text{scene}}$ on the dataset $D$; $M_{\text{rand}}$ is a random chance action classification accuracy on the dataset $D$. With this definition, we can now measure the scene representation bias of the UCF-101 [50] dataset by computing the log ratio between two accuracies: 1) the action classification accuracy of ResNet-50 backbone pre-trained on the Places365 dataset [73] on UCF-101: 59.7%. 2) the random chance accuracy on UCF-101: 1.0%. To get the first accuracy, a linear classifier is trained on UCF-101 on top of the Places365 pre-trained ResNet-50 feature backbone. As an input to the linear classifier, we use a $2,048$ dimensional feature vector extracted from the penultimate layer of the ResNet-50 feature backbone. The scene representation bias of UCF-101 is $\log(59.7/1.0) = 4.09$. A completely unbiased dataset would have $\log(1.0/1.0) = 0$ scene representation bias. Thus, the UCF-101 dataset has quite a large scene representation bias.

The reason for a scene representation bias is that human activities often occur in specific scene contexts (e.g., playing basketball in a basketball court). Figure 1 provides several examples. Even though the actors in these videos are masked-out, we can still easily infer the actions of the masked-out actors by reasoning about where the scene is. As a result, training CNN models on these examples may capture the biases towards recognizing scene contexts. Such strong scene bias could make CNN models unable to generalize to unseen action classes in the same scene context and novel tasks.

In this paper, we propose a debiasing algorithm to mitigate scene bias of CNNs for action understanding tasks. Specifically, we pre-train a CNN on an action classification dataset (Mini-Kinetics-200 dataset [64] in our experiment) using the standard cross-entropy loss for the action labels. To mitigate scene representation bias, we introduce two additional losses: (1) *scene adversarial loss* that encourages a network to learn scene-invariant feature representations and (2) *human mask confusion loss* that prevents a model from predicting an action if humans are not visible in the video. We validate the proposed debiasing method by showing transfer learning results on three different activity understanding tasks: action classification, temporal action localization, and spatio-temporal action detection. Our debiased model shows the consistent performance improvement of transfer learning over the baseline model without debiasing across various tasks.

We make the following three contributions in this paper.

- We tackle a relatively under-explored problem of mitigating scene biases of CNNs for better generalization to various action understanding tasks.
- We propose two novel losses for mitigating scene biases when pre-training a CNN. We use a scene-adversarial loss to obtain scene-invariant feature representation. We use a human mask confusion loss to encourage a network to be unable to predict correct actions when there is no visual evidence.
- We demonstrate the effectiveness of our method by transferring our pre-trained model to three action understanding tasks and show consistent improvements over the baseline.

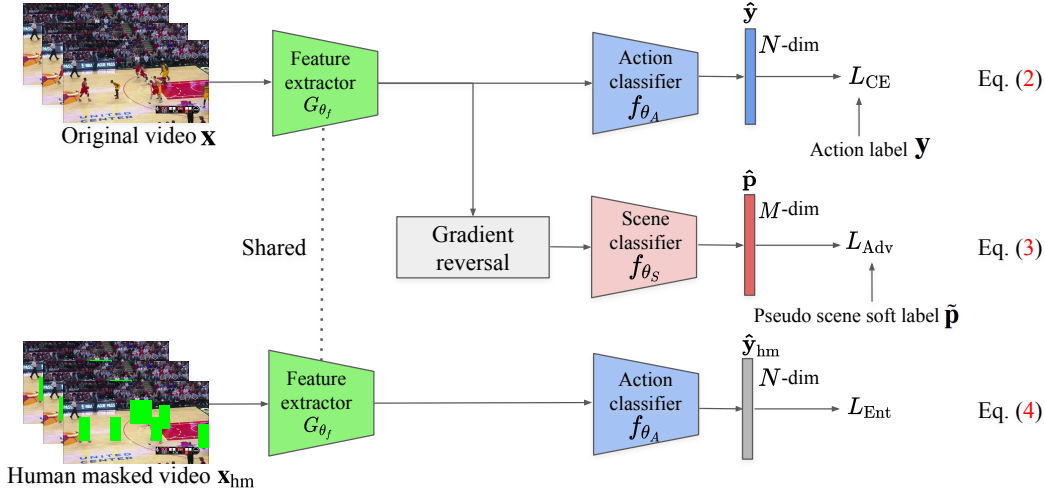

Figure 3: **Overview of the proposed approach for learning debiased video representation.** Here, our goal is to learn the parameters $\theta_f$ for the feature extractor $\phi$ by pre-training it on a large-scale video classification task. Our training involves three types of losses. First, we use a standard cross-entropy loss $L_{CE}$ for training action classification. Second, we impose a scene adversarial loss $L_{Adv}$ so that one cannot infer the scene types based on the learned representation. Third, we prepare additional videos by detecting and masking out the humans using an off-the-shelf human detector. We apply an entropy loss $L_{Ent}$ on the predicted action class distributions of the human-masked-out videos. Our intuition here is that the model should not be able to infer the correct action without seeing the evidence.

## 2    Related Work

**Mitigating biases.** Mitigating unintended biases is a critical challenge in machine learning. Examples include reducing gender or age biases for fairness [2, 5, 33, 58], easy-example biases [59, 49], and texture bias of image CNNs for improved generalization [17]. The most closely related work to ours is on mitigating scene/object/people biases by resampling the original dataset [38] to generate less biased datasets for action recognition. In contrast, we learn scene-invariant video representations from an original biased dataset *without* resampling.

**Using scene context.** Leveraging scene context is useful for object detection [40, 7, 11, 13], semantic segmentation [35, 39, 40, 69], predicting invisible things [32], and action recognition without looking at the human [23, 54]. Some work have shown that explicitly factoring human action out of context leads to improved performance in action recognition [71, 61]. In contrast to prior work that uses scene contexts to facilitate recognition, our method aims to learn representations that are invariant to scene bias. We show that the debiased model generalizes better to new datasets and tasks.

**Action recognition in video.** State-of-the-art action recognition models either use two-stream (RGB and flow) networks [46, 14] or 3D CNNs [8, 52, 64, 60, 20]. Recent advances in this field focus on capturing longer-range temporal dependencies [55, 56, 63]. Instead, our work focuses on mitigating scene bias when training these models on existing video datasets. Our debiasing approach is *model-agnostic*. We show the proposed debiasing losses improve the performance on different backbone architectures: 3D-ResNet-18 [20] and VGG-16 network [47].

**Video representation transfer for action understanding tasks.** Many recent CNNs for action classification [8, 52, 64, 60, 20], temporal action localization [66, 45, 65, 12, 70], and spatio-temporal action detection [19, 42, 41, 48, 4, 44, 62, 30, 74, 21, 26, 43] rely on pre-training a model on a large-scale dataset such as ImageNet or Kinetics and finetuning the pre-trained model on the target datasets for different tasks. While Kinetics is a large-scale video dataset, it still contains a significant scene representation bias [38]. In light of this, we propose to mitigate scene bias for pre-training a more generalizable video representation.

**Adversarial training.** Domain adversarial training introduces a discriminator to determine where the data is coming from. It has been successfully applied to unsupervised domain adaptation [16, 53] and later extended to pixel-wise adaptation [6, 24, 10] and multi-source adaptation [25]. Building upon adversarial training, recent work aims to mitigate unintended biases by using the adversary to predict the protected variables [68, 58] or quantify the statistical dependency between the inputs and the protected variables [1]. Our work uses a similar strategy for mitigating scene bias from video representation. Unlike existing work where the protected variables are given (demographic information such as gender, age, race), the ground truth scene labels of our training videos are not available. To address this, we propose to use the output of a pre-trained scene classifier as a proxy.

**Artificial occlusion.** Applying occlusion masks to input images (or features) and monitoring the changes at the output of a model has been used to visualizing whether a classifier can localize objects in an image [67], learning object detectors from weak image-level labels [3, 49, 36, 37], improving robustness of object detectors [59], and regularizing model training [18]. We adopt a similar approach by masking out humans detected by an off-the-shelf object detector. Our focus, however, differs from existing work in that we aim to train the model so that it is *unable* to predict the correct class label.

## 3  Method

### 3.1  Overview of the method

**Pre-training.** We show an overview of the proposed approach for learning debiased video representations in Figure 3. Our goal is to learn parameters $\theta_f$ of a feature extractor $G$ by pre-training it on a large-scale video classification dataset. Given a video and the corresponding action label $(\mathbf{x}, \mathbf{y}) \in \mathbf{X} \times \mathbf{Y}$, where $\mathbf{X}$ is a video dataset and $\mathbf{Y}$ is the action label set with $N$ classes, we extract features using a CNN denoted as $G_{\theta_f}$ with parameters $\theta_f$. Note that our method is *model-agnostic* in that we can use any 3D CNN or 2D CNN as our feature extractor. We feed the extracted features into an action classifier $f_{\theta_A}$ with parameters $\theta_A$. We use a standard cross-entropy loss for the action classifier for penalizing the incorrect action predictions as follows.

$$L_{CE} = -\mathbb{E}_{(\mathbf{x}, \mathbf{y}) \sim (\mathbf{X}, \mathbf{Y})} \sum_{k=1}^{N} y_k \log f_{\theta_A}(G_{\theta_f}(\mathbf{x})). \tag{2}$$

In addition to the cross-entropy loss in (2), we propose two losses for debiasing purpose: 1) scene adversarial loss $L_{\mathrm{Adv}}$, and 2) human mask confusion loss $L_{\mathrm{Ent}}$. We impose a scene adversarial loss to encourage learning scene-invariant representations. The scene adversarial loss penalizes the model if it could infer the scene types based on the learned representation. The human mask confusion loss penalizes the model if it could predict the actions when the humans in the video are masked-out. For this loss, we first detect humans in the input video and mask-out the detected humans. We then extract features of the human-masked-out video. We feed the features into the same action classifier. As shown in Figure 3 (dotted line), the weights of the feature extractor $\theta_f$ and action classifiers $\theta_A$ for the human mask confusion loss are shared with those for the action classification cross-entropy loss. We maximize the entropy of the predicted action class distributions when the input is a human-masked-out video. We provide more details on each of the two losses and the optimization process in the following subsections.

**Transfer learning.** After pre-training a CNN with the proposed debiasing method, we initialize the weights of the feature extractor, $\theta_f$ for downstream target tasks: action classification, temporal action localization, and spatio-temporal action detection. We remove the scene prediction and the human-masked action prediction heads of the network i.e., $\theta_A$, $\theta_S$. Then we finetune $\theta_f$ and the task-specific classification and regression parameters on the target datasets.

### 3.2  Scene adversarial loss

The motivation behind the scene adversarial loss is that we want to learn feature representations suitable for action classification but invariant to scene types. For example, on Figure 2 left, we aim to encourage a network to focus on a singer in the video and to predict the action as *singing*. We do not want a network to classify the video as *playing baseball* because the network recognizes that the scene is a baseball field. To enforce the scene invariance criterion to the network, we learn a scene classifier $f_{\theta_S}$ with parameters $\theta_S$ on top of the feature extractor $G_{\theta_f}$ in an adversarial fashion. We

define the scene adversarial loss as,

$$L_{\text{Adv}} = -\mathbb{E}_{(\mathbf{x},\mathbf{p}) \sim (\mathbf{X},\mathbf{P})} \sum_{m=1}^{M} p_m \log f_{\theta_S}(G_{\theta_f}(\mathbf{x})). \tag{3}$$

Here we denote a scene label as $\mathbf{p} \in \mathbf{P}$, and the number of scene types by $M$. The loss (3) is adversarial in that the parameters of the feature extractor $\theta_f$ maximize the loss while the parameters of the scene classifier $\theta_S$ minimize the loss.

**Pseudo scene label.** Most of the action recognition datasets such as Kinetics and UCF-101 do not have scene annotations. In this work, we obtain a pseudo scene label $\tilde{\mathbf{p}} \in \tilde{\mathbf{P}}$ by running Places365 dataset pre-trained ResNet-50 [73] on the Kinetics dataset.

## 3.3 Human mask confusion loss

We show the motivation for using the human mask confusion loss on the right of Figure 2. If every human, (who is *swimming* in this example), in the input video is masked out, we aim to make the network unable to infer the true action label (*swimming*). We denote a human-masked-out video as $\mathbf{x}_{\text{hm}} \in \mathbf{X}_{\text{hm}}$. We define the human mask confusion loss as an entropy function of the action label distributions as follows.

$$L_{\text{Ent}} = -\mathbb{E}_{\mathbf{x}_{\text{hm}} \sim \mathbf{x}_{\text{hm}}} \sum_{k=1}^{N} f_{\theta_A}(G_{\theta_f}(\mathbf{x}_{\text{hm}})) \log f_{\theta_A}(G_{\theta_f}(\mathbf{x}_{\text{hm}})). \tag{4}$$

Both of the parameters of the feature extractor $\theta_f$ and the action classifier $\theta_A$ maximize (4) when an input is a human-masked-out video $\mathbf{x}_{\text{hm}}$. Our intuition here is that models should not be able to predict correct action without seeing the evidence.

**Human mask.** To mask-out humans in videos, we run an off-the-shelf human detector [22] on the Kinetics dataset offline and store the detection results. During training, we load the cached human detection results. For every human bounding box in a frame, we fill in the human-mask regions with the average pixel value of the video frame, following the setting from Anne Hendricks et al. [2].

## 3.4 Optimization

Using all three losses, we define the optimization problem as follows. When an input video is an original video $\mathbf{x}$ without human masking, the optimization is

$$L(\theta_f, \theta_S, \theta_A) = L_{CE}(\theta_f, \theta_A) - \lambda L_{\text{Adv}}(\theta_f, \theta_S),$$
$$(\theta_f^*, \theta_A^*) = \underset{\theta_f, \theta_A}{\text{argmin}}\, L(\theta_f, \theta_S^*, \theta_A),$$
$$\theta_S^* = \underset{\theta_S}{\text{argmax}}\, L(\theta_f^*, \theta_S, \theta_A^*). \tag{5}$$

Here $\lambda$ is a hyperparameter for controlling the strength of the scene adversarial loss. We use the gradient reversal layer for adversarial training [15]. When an input video is a human-masked-out video $\mathbf{x}_{\text{hm}}$, the optimization is

$$(\theta_f^*, \theta_A^*) = \underset{\theta_f, \theta_A}{\text{argmax}}\, L_{\text{Ent}}(\theta_f, \theta_A). \tag{6}$$

For every iteration, we alternate between the optimization (5) and (6).

# 4 Experimental Results

We start with describing the datasets used in our experiments (Section 4.1) and implementation details (Section 4.2). We then address the following questions: i) Does the proposed debiasing method mitigate scene representation bias? (Section 4.3) ii) Can debiasing improve generalization to other tasks? (Section 4.4, Section 4.5) iii) What is the effect of the two proposed losses designed for mitigating scene bias? What is the effect of using different types of pseudo scene labels? (Section 4.6)

### 4.1 Datasets

**Pre-training.** We pre-train our model on the Mini-Kinetics-200 dataset [64]. Mini-Kinetics-200 is a subset of the full Kinetics-400 dataset [31]. Since the full Kinetics is very large and we do not have sufficient computational resources, we resort to using Mini-Kinetics-200 for pre-training a model. The training set consists of 80K videos and the validation set consists of 5K videos.[1] To validate whether our proposed debiasing method improves generalization, we pre-train models with debiasing and another model without debiasing. We then compare the transfer learning performances of the two models on three target tasks: 1) action classification, 2) temporal action localization, 3) spatio-temporal action detection.

**Action classification.** For the action classification task, we evaluate the transfer learning performance on the UCF-101 [50], HMDB-51 [34], and Diving48 [38] datasets. UCF-101 consists of 13,320 videos with 101 action classes. HMDB-51 consists of 6,766 videos with 51 action classes. Diving48 [38] is an interesting dataset as it contains no significant biases towards the scene, object, and human. Diving48 consists of 18K videos with 48 fine-grained diving action classes. We use the train/test split provided by Li et al. [38]. Videos in all three datasets were temporally trimmed. We report top-1 accuracy on all thee splits of the UCF-101 and the HMDB-51 datasets. The Diving48 dataset provides only one split. Thus we report the top-1 accuracy on this split.

**Temporal action localization.** Temporal action localization is a task to not only predict action labels but also localize the start and end time of the actions. We use the THUMOS-14 [29] dataset as our testbed. THUMOS-14 contains 20 action classes. We follow Xu et al. [65]'s setting for training and testing. We train our model on the temporally annotated validation set with 200 videos. We test our model on the test set consisting of 213 videos. We report the video mean average precision at various IoU threshold values.

**Spatio-temporal action detection.** Given an untrimmed video, the task of spatio-temporal action detection aims to predict action label(s) and also localize the person(s) performing the action in both space and time (e.g., an action tube). We use the standard JHMDB [27] dataset for this task. JHMDB is a subset of HMDB-51. It consists of 928 videos with 21 action categories with frame-level bounding box annotations. We evaluate models on all three splits of JHMDB. We report the frame mean average precision at the IoU threshold 0.5 as our evaluation metric.

### 4.2 Implementation details

We implement our method with PyTorch (version 0.4.1). We choose 3D-ResNet-18 [20] as our feature backbone for Mini-Kinetics-200 → UCF-101/HMDB51/Diving48, and Mini-Kinetics-200 → THUMOS-14 experiments because open-source implementations for action classification [20] and temporal action localization [57] are available online. We use a 3 channels × 16 frames × 112 pixels × 112 pixels clip as our input to the 3D-ResNet-18 model. We use the last activations of the `Conv5` block of the 3D-ResNet-18 as our feature representation $G_{\theta_f}(\mathbf{x})$.

For the spatio-temporal action detection task, we adopt the frame-level action detection code [48] based on the VGG-16 network [47]. We use a 3 channels × 1 frame × 300 pixels × 300 pixels frame as our input to the VGG-16 network. We use the `fc7` activations of the VGG-16 network as our feature representation $G_{\theta_f}(\mathbf{x})$.

We use a four-layer MLP as our scene classifier, where the hidden fully connected layers have 512 units each. We choose $\lambda = 0.5$ for the gradient reversal layer [15] using cross-validation. We set the batch size as 32 with two P100 GPUs. We use SGD with a momentum of 0.9 as our optimizer. We set the weight decay as 0.00001. The learning rate starts from 0.001, and we divide it by 10 whenever the validation loss saturates. We train our network for 100 epochs. We use the validation loss on Mini-Kinetics-200 for model selection and hyperparameter tuning. When we conduct the transfer learning on the target datasets of the target tasks, we follow the same hyperparameter settings of Hara et al. [20], Wang and Cheng [57] and Singh et al. [48] for action classification, temporal action localization, and spatio-temporal action detection, respectively.

Table 1: Transfer learning results on action classification. The video representation trained using the proposed debiasing techniques consistently improves the accuracy on new datasets. For HMDB-51 and UCF-101, we show the average accuracy of all three test splits. All methods but TSN use a clip length of 16. In the first block, we list the accuracies of the other methods.

| Method | Backbone | HMDB-51 | UCF-101 | Diving48 |
|---|---|---|---|---|
| C3D [61] | C3D [51] | - | 82.3 | - |
| Factor-C3D [61] | C3D [51] | - | 84.5 | - |
| RESOUND-C3D [38] | C3D [28] | - | - | 16.4 |
| TSN [55] | BN-Inception | 51.0 | 85.1 | 16.8 |
| 3D-ResNet-18 [20] | 3D-ResNet-18 | 53.6 | 83.5 | 18.0 |
| 3D-ResNet-18 [20] + debiased (ours) | 3D-ResNet-18 | 56.7 | 84.5 | 20.5 |

## 4.3 Scene classification accuracy

On the Mini-Kinetics-200 validation set, our scene classifier achieves an accuracy of 29.7% when training the action classification without debiasing (i.e., , with standard cross-entropy loss only).[2] With debiasing, the scene classification accuracy drops to 2.9% (the accuracy of random guess is 0.3%.) The proposed debiasing method significantly reduces scene-dependent features.

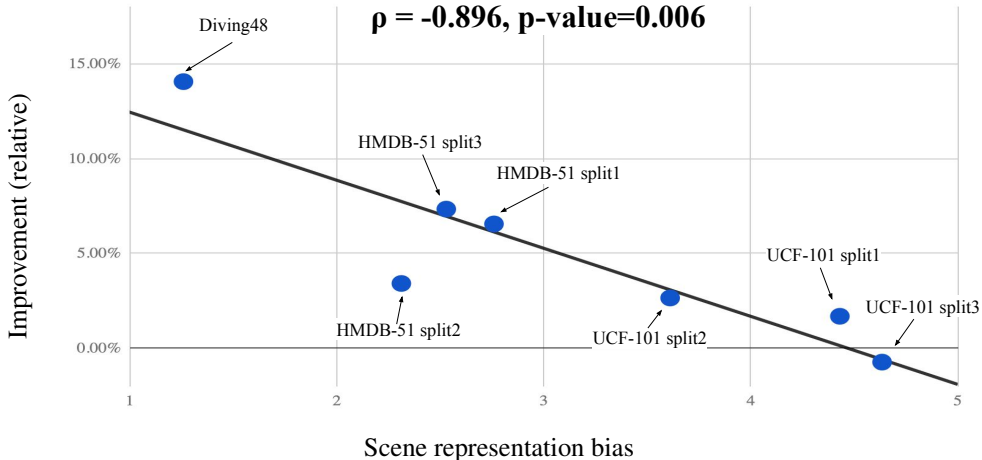

Figure 4: **A strong negative correlation between relative performance improvement and scene representation bias.** We measure the scene representation bias defined as (1) and the relative improvement of each split of the HMDB-51, UCF-101, Diving48 datasets between models trained without and with the proposed debiasing method. The Pearson correlation is $\rho = -0.896$ with a $p$-value 0.006, highlighting a *strong negative correlation*.

## 4.4 Transfer learning for action classification

Table 1 shows the results on transfer learning for action classification on other datasets. As shown in the last two rows of Table 1, our debiased model consistently outperforms the baseline without debiasing on all three datasets. The results validate that mitigating scene bias can improve generalization to the target action classification datasets. As shown in the first block of Table 1, action-context factorized C3D (referred to as Factor-C3D) [61] also improves the baseline C3D [51] on UCF-101. The accuracy of Factor-C3D is on par with our debiased 3D-ResNet-18. Note that the model used in Factor-C3D is 2.4× larger than ours. We also compare our method with RESOUND-C3D [38] on the Diving48 dataset. All the videos in the Diving48 dataset share similar scenes. Our proposed debiasing method shows a favorable result compared to [38]. We show a large relative improvement (14.1%) on the Diving48 dataset since it has a small scene representation bias of 1.26 [38].

Table 2: Transfer learning results on temporal action localization task: THUMOS-14 dataset. The evaluation metric is the video mAP at various IoU threshold values. The video representation trained using the proposed debiasing techniques consistently improves the performance on the new task. In the first block, we list the mAPs of the state-of-the-arts.

| Method | Inputs | Backbone | IoU threshold | | | | | | | |
| --- | --- | --- | --- | --- | --- | --- | --- | --- | --- | --- |
| | | | 0.1 | 0.2 | 0.3 | 0.4 | 0.5 | 0.6 | 0.7 | avg. |
| TAL-Net [9] | RGB+Flow | I3D | 59.8 | 57.1 | 53.2 | 48.5 | 42.8 | 33.8 | 20.8 | 45.1 |
| SSN [70] | RGB+Flow | InceptionV3 | 66.0 | 59.4 | 51.9 | 41.0 | 29.8 | 19.6 | 10.7 | 39.8 |
| R-C3D [65] | RGB | C3D | 54.5 | 51.5 | 44.8 | 35.6 | 28.9 | - | - | - |
| CDC [45] | RGB | C3D | - | - | 40.1 | 29.4 | 23.3 | 13.1 | 7.9 | - |
| R-C3D [57] | RGB | 3D-ResNet-18 | 48.6 | 48.6 | 45.6 | 40.8 | 32.5 | 25.5 | 15.5 | 36.7 |
| R-C3D [57] + debiased (ours) | RGB | 3D-ResNet-18 | 50.2 | 50.5 | 47.9 | 42.3 | 33.4 | 26.3 | 16.8 | 38.2 |

Table 3: Transfer learning results on spatio-temporal action detection. The evaluation metric is the frame mAP at the IoU threshold of 0.5. We list the frame mAP of the state-of-the-arts for reference.

| Method | Backbone | Inputs | Pre-train | JHMDB (all splits) |
| --- | --- | --- | --- | --- |
| ACT [30] | VGG | RGB+Flow | ImageNet | 65.7 |
| S3D-G [64] | Inception (2D+1D) | RGB+Flow | ImageNet+FullKinetics | 75.2 |
| ROAD [48] | VGG | RGB | ImageNet+MiniKinetics | 32.5 |
| ROAD [48] + debiased (ours) | VGG | RGB | ImageNet+MiniKinetics | 34.5 |

**Relative performance improvement vs. scene representation bias.** Figure 4 illustrates the relationship between relative performance improvement from the proposed debiasing method and the scene representation bias (defined in [38]). We measure the scene representation bias defined as (1) and the relative improvement of each split of the HMDB-51, UCF-101, Diving48 datasets. The Pearson correlation is $\rho = -0.896$ with a $p$-value 0.006, highlighting a *strong negative correlation* between the relative performance improvement and the scene representation bias. Our results show that if a model is pre-trained with debiasing, the model generalizes better to the datasets with *less* scene bias, as the model pays attention to the actual action. In contrast, if a model is pre-trained on a dataset with a significant scene bias e.g., Kinetics without any debiasing, the model would be biased towards certain scene context. Such a model may still work well on target dataset with strong scene biases (e.g., UCF-101), but does not generalize well to other *less* biased target datasets (e.g., Diving48 and HMDB-51).

## 4.5   Transfer learning for other activity understanding tasks.

**Temporal action localization** Table 2 shows the results of temporal action localization on the THUMOS-14 dataset. Using the pre-trained model with the proposed debiasing method consistently outperforms the baseline (pre-training without debiasing) on all the IoU threshold values. Our result suggests that a model focusing on the actual discriminative cues from the actor(s) helps localize the action.

**Spatio-temporal action detection** Table 3 shows spatio-temporal action detection results. With debiasing, our model outperforms the baseline without debiasing on the JHMDB dataset. The results in Table 2 and 3 validate that mitigating scene bias effectively improves the generalization of pre-trained video representation to other activity understanding tasks.

## 4.6   Ablation study

We conduct ablation studies to justify the design choices of the proposed debiasing technique. Here we use the Mini-Kinetics-200 → HMDB-51 setting for the ablation study.

**Effect of the different pseudo scene labels.** First, we study the effect of pseudo scene labels for debiasing in  Table 4. Compared to the model *without* using scene adversarial debiasing, using hard pseudo labels improves transfer learning performance. Using soft pseudo labels for debiasing further enhances the performance. We attribute the performance improvement to many semantically similar scene categories in the Places365 dataset. Using soft scene labels alleviates the issues of committing to one particular scene class. In all the remaining experiments, we use pseudo scene soft labels for scene adversarial debiasing.

Table 4: Effect of debiasing using different pseudo scene labels generated from the off-the-shelf classifier.

| Pseudo label used | HMDB-51 | | | |
| | split-1 | split-2 | split-3 | avg. |
| --- | --- | --- | --- | --- |
| None (w/o debiasing) | 52.9 | 55.4 | 52.6 | 53.6 |
| Hard label | 54.8 | 54.2 | 54.6 | 54.5 |
| Soft label (ours) | **56.4** | **55.9** | **56.4** | **56.2** |

Table 5: Effect of using different losses for reducing scene bias. Both losses improve the performance in transfer learning.

| Loss | | HMDB-51 | | | |
| $L_{Adv}$ | $L_{Ent}$ | split-1 | split-2 | split-3 | avg. |
| --- | --- | --- | --- | --- | --- |
| × | × | 52.9 | 55.4 | 52.6 | 53.6 |
| × | ✓ | 55.0 | 55.3 | 55.1 | 55.1 |
| ✓ | × | **56.4** | 55.9 | 56.4 | 56.2 |
| ✓ | ✓ | **56.4** | **57.3** | **56.5** | **56.7** |

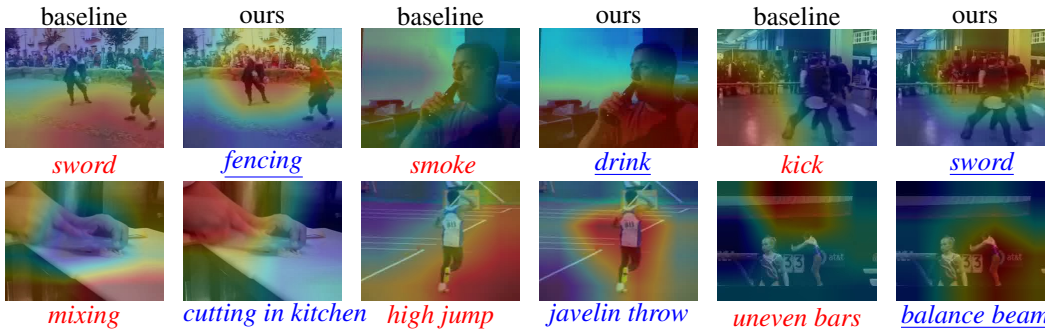

| baseline | ours | baseline | ours | baseline | ours |
| --- | --- | --- | --- | --- | --- |
| *sword* | *fencing* | *smoke* | *drink* | *kick* | *sword* |
| *mixing* | *cutting in kitchen* | *high jump* | *javelin throw* | *uneven bars* | *balance beam* |

Figure 5: **Class activation maps (CAM) on the HMDB-51 (first row) and UCF-101 (second row) datasets.** The words underlined in blue are correct predictions, and those in red with no underline are incorrect predictions. The video representation using the proposed debiasing algorithm focuses more on the direct visual cues (i.e., the main actors) rather than the surrounding scene contexts.

**Effect of the two different losses.** Next, we ablate each of the two debiasing loss terms: scene adversarial loss ($L_{Adv}$) and human mask confusion loss ($L_{Ent}$) in Table 5. We observe that both losses improve the performance individually. Using both debiasing losses gives the best results and suggests that the two losses are regularizing the network in a complementary fashion.

## 4.7 Class activation map visualization

To further demonstrate the efficacy of our debiasing algorithm, we show class activation maps (CAM) [72] from models with and without debiasing in Figure 5. We show the CAM overlaid over the center frame (the eighth frame out of the sixteen frames) of each input clip. We present the results on the HMDB-51 and UCF-101 datasets. We observe that without debiasing, a model predicts the incorrect classes because the model focuses on the scene instead of human actors. However, with debiasing, a model focuses on human actions and predicts the correct action classes.

## 5 Conclusions

We address the problem of learning video representation while mitigating scene bias. We augment the standard cross-entropy loss for action classification with two additional losses. The first one is an adversarial loss for scene class. The second one is an entropy loss for videos where humans are masked out. Training with the two proposed losses encourages a network to focus on the actual action. We demonstrate the effectiveness of our method by transferring the pre-trained model to three target tasks.

As we build our model upon relatively weak baseline models, our model's final performance still falls behind other state-of-the-art models. In this work, We only addressed one type of representation bias, i.e., scene bias. Extending the proposed debiasing method to mitigate other kinds of biases e.g., objects and persons for human action understanding is an interesting and important future work.

**Acknowledgment.** This work was supported in part by NSF under Grant No. 1755785 and a Google Faculty Research Award. We thank the support of NVIDIA Corporation with the GPU donation.

## Footnotes

[1]As the sources of Kinetics dataset are from YouTube videos, some of the videos are no longer available. The exact number of training videos we used is 76,103, and the number of validation videos is 4,839.

[2]Since there are no ground truth scene labels in the Mini-Kinetics-200 dataset, we use pseudo labels to measure the scene classification accuracy.

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
