[Reviews · NeurIPS 2019]

Reviewer 1



Strengths of the paper are listed as follows: S1. The paper tackles the important problem of scene de-biasing for action recognition. It is of high concern for computer vision community to sanity check whether the proposed models (really) learn the dynamics of actions, and not just learn to leverage spurious bias such as the co-occurrence of the scene between actions. S2. The authors develop a sensible solution, forcing the model to consider the human region for recognition, trying to reduce the sensitivity of action representation to the surrounding context. This is achieved by borrowing ideas from adversarial learning, that is, the scene recognition ability of action code is altered by directly using gradient reversal [8], a well-known domain confusion method in the literature since 2015. S3. The authors conduct several action-related experiments to showcase the ability of their models. Weaknesses of the paper are listed as follows: W1. (Problem-wise weakness): The first concern about the paper is theoretical. This paper frames the surround 'scene' as a 'degenerative bias' that leads to misclassification. Where we agree that scene bias is evident in action recognition datasets (due to sampling bias, see Computer vision meets fairness talk, https://visualai.princeton.edu/slides/Fairness_CVPR2018.pdf), many actions have a preferred scene: Diving and Swimming happens in the pool whereas Baseball happens in the stadium. Action-Scene duality is highly visible in the visual world. So, in the end, any form of action recognition datasets will exhibit such bias, which can carry useful supplemental information about the action. This raises the following important questions: - Should we really learn to ignore the surround (and be invariant to it, as is done in this paper)? - Or learn to adaptively use it only when it is correlated with the scene (more in disentangling sense, see for example "Recognize Actions by Disentangling Components of Dynamics", CVPR 2018)? - Or should we designate tasks that happen in the same environment, but only the action class changes (see, for example, The 20BN-SOMETHING-SOMETHING dataset, which has been shown to be not solvable without considering human-object appearance)? In this manner, the paper is missing a discussion on these aspects of the action-scene duality. The quantitative results presented in the paper may tell us that, indeed scene should not be ignored, but should be factored out from the action, so that it is up to classifier when to rely on it or when to ignore it. In this respect, the authors are missing an important comparison to a highly relevant paper ("Pulling Actions out of Context: Explicit Separation for Effective Combination", Wang and Hoai, CVPR 2018) that factors out the action from the surround context, by masking out the human region, as is done by this submission. See experimental weaknesses for detailed comments. W2. (Methodological weakness): W.2.1. The developed method in this paper is borrowed from (Ganin and Lempitsky, ICML 2015) without modification. (Ganin and Lempitsky, ICML 2015) proposes a domain confusion objective, where the bottleneck network is trained to confuse from which domain the input is coming from. The authors adopt the same idea, whereas the source of confusion is the scene factor instead of the domain. The authors fairly cite this paper. Although this limits methodological novelty in this work. W.2.2. A similar objective for the second part of the loss, that strives for negative correlation of scene-only input, where the human is masked out is explored by (Wang and Hoai, CVPR 2018). In that paper, the authors utilize masked input to factorize the contextual factors from the action code, whereas in this paper it is used for de-correlation purposes only. W.2.3. A bounding box is a rough localization of a non-rigid object like human. This means more than 60% of pixels within the bounding box still belongs to the surround scene, and leaks into the action code. This limits the use of the method when there is a high co-occurrence of surround pixels within the bounding box. W3. (Experimental weakness): W.3.1. No comparison to (Wang and Hoai, CVPR 2018). Reducing the effect of the context in the action code for video action recognition has been previously explored in (Wang and Hoai, CVPR 2018), using C3D and I3D backbone architectures, using the same set of datasets. This makes (Wang and Hoai, CVPR 2018) a natural competitor for the proposed solution. However, the authors neither mention nor compare their solution to this highly relevant paper. W.3.2. No comparison to (Li et al, ECCV 2018, RESOUND: Towards Action Recognition without Representation Bias). This paper tackles dataset bias in action recognition, not limited to the scene. Even though the authors list this paper as relevant, there is no quantitative comparison against their approach either. This paper also proposes a new dataset called Diving48, which is a fine-grained action recognition dataset, and a natural data for the authors to showcase the ability of their model of ignoring the surround information (and being solely focusing on the actor). W.3.3. No evaluation of (altered) scene recognition performance provided. One of the objectives that the authors use enforces the model to misclassify the scene from the action code. However, the ability of this objective to reduce sensitivity for scene variability is only indirectly evaluated (through action recognition performance). The authors do not show that indeed using this objective, the action code performs poorly in scene recognition (as provided by Places-365 Resnet-50 model) as opposed to the vanilla action code that is learned without adversarial objective (e.g., I3D). In that sense, we don't have a good idea of to what extent the objective is met, to what extent indeed vanilla I3D or C3D exhibits scene information (Although, since there is no ground truth scene information provided, the results should be read in this manner). W.3.4. The choice of baseline models for de-biasing is unjustified. The authors choose to de-bias 3D-Resnet-18, R-C3D, and ROAD for de-biasing for three aforementioned tasks. As listed in the Tables, none of these models are current state-of-the-art for the considered datasets. In this manner, the authors choose to de-bias a low performing model for three tasks, leading to inferior results against the best model. This raises the question: Is the proposed solution not suitable for better performing models? Or Is scene-bias more severe in inferior models, making it infeasible to apply to the current state-of-the-art ? How would the performance improvement pronounce for I3D or S3D ? W.3.4. Obtained improvement on three tasks is insignificant. The authors try to justify the low and lower improvement in different datasets and different splits via the amount of scene bias measured by (Li et al, 2018). However, a concrete scatter plot which plots the obtained improvement (if any) against the amount of scene bias within these datasets is not provided. In this manner, it is almost impossible to judge where the improvement is coming from. W4. (Related work weakness, Minor): W.4.1. Severe scene bias of UCF-101 has been recognized before by (He et al, 2016, Human Action Recognition without Human) where the authors classify actions by masking out human. The paper may cite this as a relevant work.

Reviewer 2



The main weakness of the paper is its relatively low experimental results. It seems the proposed method is helping weak baselines, however, results are quite below the state of the art. As actions and scenes are correlated in current datasets, one way to demonstrate the advantage of the approach could be action classification and detection in untypical scenes. It would be good to see the evidence that the method improves strong baselines in such settings.

Reviewer 3



This work investigates an open problem in action recognition, namely the strong correlation between action and scene. Such a correlation can lead to an overfitting to scene features over action-specific features, which in turn can lead to reduced generalization and recognition out of context. The proposed method makes sense, the link between actions and scenes is fairly investigated, and the approach is evaluated on multiple tasks, both to show the generalization of the method to any setting and to investigate the scene bias in multiple settings. The work does have a number of limitations: The experiments are not thoroughly convincing. The introduction, method, and figures highlight the importance of recuding scene bias. The method is however still evaluated on well-known action datasets, all with a known high scene bias. As a result, the experiments paint a picture that scene debiasing is far from important. While there is some gain for HMDB51, there is hardly any gain on UCF-101 and THUMOS-14 (temporal detection), while there is no direct gain on UCF-24 (spatio-temporal detection). The low impact of the proposed approach invariably leads to the conclusion that scene debiasing is not warranted. Figure 4 shows that the focus is now more on the actors, but the results are not really gaining from this focus. Why did the experiments focus on UCF-101, THUMOS-14, UCF-24, and HMDB51? Why not investigate a dataset with static backgrounds or with fixed scenes? There big gains are possible, because the only way to recognize the action is to look at the actor. With the current set of experiments, the only conclusion that the reader can make is that scene debiasing is intuitive but not important. Lastly a few smaller points: - Why only show results for HMDB51 in Tables 1 and 2? Is that because UCF-101 does not show good improvements? - What is the reason for not including L_{ENT} for spatio-temporal detection? - Why is scene debiasing important for (spatio-)temporal detection compared to classification? - Throughout the text, there are a number of typos and awkward sentences, e.g. lines 27 and 163.

[Author Response · NeurIPS 2019]

**R1,R2,R3:** *Improvement and scene bias:* We show a scatter plot on the relative improvement and scene representation bias of target datasets on the figure on the right side (best viewed with zoom). We measure the scene bias defined by Li et al.(referred to as "RESOUND"). We show the scene bias and the relative improvement of each split of the HMDB-51, UCF-101, Diving48 datasets. The Pearson correlation is $\rho = -0.896$ with a *p*-value 0.006, highlighting a strong negative correlation between the relative improvement and the scene bias.

**R1,R2,R3:** *Results on Diving48:* We compare our method with RESOUND C3D (L=16) on the Diving48 dataset. All the videos in the Diving48 dataset share similar scenes. Our proposed debiasing method shows a favorable result compared to RESOUND. We show a relatively larger relative improvement (14.1%) on the Diving48 dataset since it has a small scene bias of 1.26.

**R1:** *Action-scene factorization vs. scene-invariance action features:* We believe both directions are worth pursuing. Both action-scene factorization and scene-invariant action features are effective in *typical* scenarios. For example, basketball players play basketballs in a basketball court. However, in *atypical* scenarios, learning a scene-invariant model is particularly important. For example, a singer is singing a song in a baseball field. Factorized models may incorrectly predict the actions because factorized models rely on the scene as well. On the other hand, scene-invariant models focus on the actual action taking place. Consequently, they can perform well, particularly for out-of-context actions. We will cite the missing papers and discuss the scene-action duality in the revision.

|  | Ours | RESOUND |
|---|---|---|
| w/o. debiasing | 18.0 | 16.4 |
| w. debiasing | 20.5 | N/A |

**R1:** *Methodology:* Our method differs from Ganin and Lempitsky, in that we consider another objective: *human mask entropy*. Specifically, we maximize the entropy of our model's action prediction when humans are masked out in the video. Our method differs from Wang and Hoai (referred to as "Factor") because we mask out humans by explicitly detecting them, while Factor temporally masks out actions (not humans) in the video by using conjugate examples. Note that there is no guarantee that the conjugate examples corresponding to the current action examples do not contain the actions, especially when the dataset is temporally trimmed e.g., UCF-101 and HMDB-51.

**R1:** *Information from the rest of the pixels (60%) within bounding boxes could leak into the action code?:* The information from the rest of the pixels (60%) within bounding boxes does not leak into the action code. Our training objective is to maximize the entropy of a model prediction, $L_{\text{Ent}}$, when detected bounding boxes are *masked out*. In this case, a model sees only the background context but not the actor. Maximizing the entropy of action prediction, therefore, prevents the background information from leaking into the action code.

**R1:** *Comparison with Factor:* We compare ours with Factor on the UCF-101 dataset. They show results on the Hollywood2 dataset on which we do not conduct experiments (instead we have results on HMDB-51). Note that the model

|  | Ours | F-C3D | F-DTD | F-EigenTSN |
|---|---|---|---|---|
| Original | 83.5 | 82.2 | 90.8 | 95.8 |
| Proposed | 84.5 | 84.5 | 91.3 | 95.8 |

used in Factor C3D (F-C3D) is 2.4× larger than ours. A comparison with Factor DTD+C3D (F-DTD) and Factor EigenTSN+C3D (F-EigenTSN) is not fair as they are both ensemble methods while ours is not.

**R1, R2:** *Weak baselines:* Unfortunately, we do not have access to computational resources for training deeper 3D networks such as 3D-ResNet-151 or I3D. We thus resort to using lighter backbones (3D-ResNet-18 and VGG-16). However, this does not undermine our core novelty as we focus on the improvement induced by our debiasing method. Here, we show the results using a deeper 3D-ResNet-50 backbone. On HMDB-51 dataset, without debiasing, the accuracy is 59.6%, and with debiasing, the accuracy is 60.1%. We also plan to plug-in our debiasing to the state-of-the-art backbone models in the future.

**R1:** *Scene classification accuracy:* On the Mini-Kinetics-200 validation set, without debiasing, the scene classification accuracy is 29.7%. With debiasing, the scene classification accuracy drops to 2.9%. The random chance is 0.3%. The proposed debiasing method indeed reduces the scene-dependent feature representation. Please note that there are no ground truth scene labels. Here we use pseudo labels to measure the scene classification accuracy.

**R1,R3:** *Missing citations, typos, and awkward sentences:* We will cite the missing related work (Factor, Zhao et al., He et al., Vu et al., and Khosla et al.) and discuss them in the revision. We will revise the paper accordingly.

**R3:** *Why not using $L_{\text{Ent}}$ for spatio-temporal detection?, Why ablation only on HMDB-51:* We show the result of using both $L_{\text{Adv}}$ and $L_{\text{Ent}}$ in the spatio-temporal action detection experiment in the table on the right side. Adding $L_{\text{Ent}}$ improves the performance by 0.1 point. Due to the limited resources, we conducted the ablation experiments on the relatively smaller HMDB-51 dataset for the classification task.

| $L_{\text{Adv}}$ | $L_{\text{Ent}}$ | frame mAP |
|---|---|---|
| × | × | 32.5 |
| ✓ | × | 34.4 |
| ✓ | ✓ | 34.5 |

**R3:** *Why debiasing is important for detection compared to classification:* We can imagine a scenario that an actor is running and the background scene is changing from urban to suburb. Debiased models can (spatio-)temporally localize the action correctly by focusing on the actual action. However, a non-debiased model, which is trained on the dataset where running only happens with the urban scene, might (spatio-)temporally localize the running action only in the urban scene and fail to recognize the running in the suburb scene.

[Meta-Review · NeurIPS 2019]

The initial scores for this paper were : 4: An okay submission, but not good enough; a reject. 7: A good submission; an accept. 5: Marginally below the acceptance threshold. The main concerns of the negative reviewers were: - issues about the problem formulation - only weak baselines are considered; results below the state-of-the-art - limited novelty - missing citations - only relatively minor improvements obtained by the proposed approach The positive reviewer also acknowledges the issues with experimental evaluation (the proposed method is shown to help weak baselines that are overall below the state-of-the-art), but finds the idea of the paper interesting, original and standing out. The authors provide a rebuttal. In the follow-up discussion among the reviewers, R3 acknowledges that some of their concerns have been addressed but remains borderline negative (5) as they think the rebuttal does not alleviate the concerns regarding the overall low results and some ablations are still missing. R2 agrees on the issues with experimental evaluation pointed by R1+R3 but maintains that “given that the problem and the method are interesting and that there are no good dataset to study them, I would recommend accept.” R1 is only partially happy with the rebuttal but agrees with R2 that the current field of action recognition needs new ideas, current datasets are biased and far from reality. R1 is borderline about this work. 
AC has read the reviews, rebuttal and the paper. AC agrees with R2 that the problem and method are interesting. AC also agrees with R1 and R3 (and R2) on the main concern that the paper currently shows only improvements over non-state-of-the-art baselines. AC specially appreciates the additional results in the rebuttal that show new results on a additional dataset that has the same scene background (Diving48 dataset) as requested and also appreciated by R3. The proposed method demonstrates an improvement on this data and a favourable comparison to the comparable (though non-state-of-the-art) baseline. Overall, AC agrees with R2 that the paper studies and interesting and important problem and proposes an interesting solution. AC further thinks that, given the additional results in the rebuttal on the Diving48 dataset, the paper sufficiently demonstrates the potential of the proposed technique. The fact that the implementation of the method presented in this paper does not build on the state-of-the-art backbone architectures and hence does not demonstrate improvements over state-of-the-art results is a weakness. However, AC agrees with R2 that the problem and solution are interesting and this outweighs the short-coming of non-state-of-the-art results.